# Study on the Common Rail Type Injector Nozzle Design Based on the Nozzle Flow Model

**Sang-Wook Han [1], Yun-Sub Shin [1], Hyun-Chul Kim [1] and Gee-Soo Lee [2],\*** 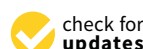

[1] Manufacturing Technology R&D Center, Korea Automotive Technology Institute, Cheonan-si 31214, Korea; swhan@katech.re.kr (S.-W.H.); yssin@katech.re.kr (Y.-S.S.); hckim@katech.re.kr (H.-C.K.)
[2] Department of Automotive Engineering, Tongmyong University, Busan 48520, Korea
\* Correspondence: gslee@tu.ac.kr; Tel.: +82-51-629-1265

**Abstract:** In this paper, a nozzle flow model was used to design an injector nozzle and obtain initial spray conditions for the dimethyl ether (DME) common rail-injection system. In order to deliver the same amount of energy as that provided by diesel at a low injection pressure of 50 MPa, the injector for DME needs nozzle holes with larger diameters and a higher SAC volume for the same injection duration. In addition, the needle lift and needle seat diameter should be increased to maintain a minimum flow area ratio. Although the vapour pressure and maximum injection pressure of DME are lower than those of diesel, the nozzle in a DME system showed higher discharge coefficients and effective nozzle exit diameters for the same injection duration owing to low kinematic viscosity. However, because the maximum injection pressure in DME is lower than that with diesel, and the length of the cavitation region is narrower.

**Keywords:** nozzle flow model; DME (dimethly ether); injector; common-rail injection system

## 1. Introduction

Dimethyl ether (DME) is a type of ether compound that combines one oxygen molecule and two methyl radicals. It is obtained by dehydration of methanol at low temperatures. DME is characterized by a relatively high cetane number and can be used as an alternative fuel for compression ignition engines. In addition, DME produces very little particulate matter because it has a high oxygen content and the absence of C-C (carbon to carbon) bonds in the molecular structure [1,2]. It started receiving attention as an alternative fuel for diesel in the 1990s as it has the potential to utilize the existing LPG (liquefied petroleum gas) transport infrastructure owing to similar properties [3–6].

A number of studies have been conducted to replace diesel fuel with DME; they have aimed to optimize engine performance and offer applications in vehicles. In previous investigations [7–9], it has been demonstrated that NOx (nitrogen oxide) and PM (particulate matter) emissions and combustion noise from compression ignition engines were reduced by adopting DME compared to diesel. Yang et al. [7] investigated the effects of injection pressure and injection rate on DME and diesel engines performance and emission characteristics. Exhaust gas characteristics were improved by varying the pilot injection period and needle lift. In addition, more research is being done on improving and controlling the fuel injection system to overcome the problem of differences in fuel properties and low heating value of the fuel itself. The viscosity of DME is lower than that of diesel fuel, causing leakage from the fuel supply system. Its lower lubricity characteristics can cause intensified surface wear of moving parts within the fuel injection system. Therefore, the maximum injection pressure of DME was found to be limited and lower than that of diesel [5].

When DME is applied to conventional diesel common rail injectors, the speed of the needle lift is slower because DME has a relatively higher compressibility. Due to the large loss of pressure in

the nozzle seat, the speed at the exit of the nozzle hole is reduced. There is also a high possibility of pulsation between the fuel pump and the injector [6]. In order to compensate for the amount of heat generated, nozzle hole diameters are basically larger with the conventional diesel common rail injectors. DME ($Q_{LHV}$) was found to have a lower heating value of 27.6 (MJ/kg), 65% of that of the diesel; a larger amount of DME is needed to supply the combustion chamber and obtain the equivalent engine power. In addition, the minimum pass-through flow area of the needle seat should also be considered, since nozzle hole size increase alone does not provide the desired flow rate [10].

In this study, the nozzle for the DME common rail injector with $P_{max} = 50$ (MPa) was designed as such to achieve the same power output as conventional diesel engines with $P_{max} = 1600$ (MPa) using a zero-dimensional nozzle flow model for the same injection duration. The flow condition of the nozzle exit affects the vaporization of fuel, directly related to the combustion efficiency and the engine power. The peak injection rate, the discharge coefficient, and the effective diameter of the nozzle exit were analyzed for injection pressure and nozzle geometry for diesel and DME. The peak injection rate based on the Bosch tube method was measured to validate the nozzle flow model. The new injector nozzle with $\varnothing = 0.245$ (mm), $h = 0.15$ (mm) and $D_{st} = 1.88$ (mm) for DME injection pressure of 50 (MPa) can obtain the same engine power as that of the diesel common rail system.

## 2. Methodology

### 2.1. Details of Injector

Figure 1 shows a schematic diagram and SEM (Scanning Electron Microscope) image of an injector nozzle. The injector (solenoid driven, maximum fuel injection pressure of 160 MPa) to be modified in this study has a Mini-SAC nozzle (conical sac hole with conical tip). The base injector nozzle has $\varnothing = 0.14$ (mm) and 8 holes, and the injection angle, which is the angle between the injected fuel spray, is 152° (see Figure 1a). The diameter of the nozzle hole along the nozzle length is constant. In Figure 1a, the fuel in the injector is sealed through line contact of the inner nozzle surface and needle seat, and the minimum pass-through flow area of the seat ($A_{th}$) is the cross-sectional area, which has been calculated as the area corresponding to the normal distance from the needle seat during needle behaviour at the line contact point. The nozzle flow model is based on the idea that the flow in the nozzle is modelled as a quasi-steady, zero-dimensional flow, and the flow can be divided and analysed for cavitating flow and non-cavitating flow, as shown in Figure 1b [11,12]. In general, the nozzle flow model is useful for determining boundary conditions for the spray analysis and designing the injection flow rate in the nozzle; it is possible to predict these conditions based on experimental data. In this study, the proposed nozzle flow model was used to design the nozzle concept for different flow regimes and nozzle geometries without experimental data. However, more accurate injection rate characteristics require full modeling and analysis, including flow passages, hydraulic valves, and actuator in the injector body. Specifications for the nozzle and needle and material properties are summarized in Tables 1 and 2, respectively.

**Table 1.** Specifications of the base injector nozzle.

| Item | Unit | Value |
|------|------|-------|
| Number of hole ($N_h$) | - | 8 |
| Diameter of hole ($D$) | mm | 0.124 |
| Length of hole ($L_n$) | mm | 0.756 |
| Diameter of SAC ($D_s$) | mm | 0.44 |
| Diameter of needle seat ($D_{st}$) | mm | 0.94 |
| Seat angle ($\theta$) | ° | 60 |
| Injection angle ($\alpha$) | ° | 152 |

**Table 2.** Material properties of the diesel and DME at $T = 20^\circ$ [5].

| Item | Unit | Diesel | DME |
|---|---|---|---|
| Density ($\rho_l$) | kg/m$^3$ | 831 | 667 |
| Kinematic viscosity ($\nu$) | cSt | 3 | 0.1 |
| Lower heating value ($Q_{LHV}$) | MJ/kg | 42.5 | 27.6 |
| Vapour pressure ($p_{vap}$) | Pa | 10,000 | 530,000 |

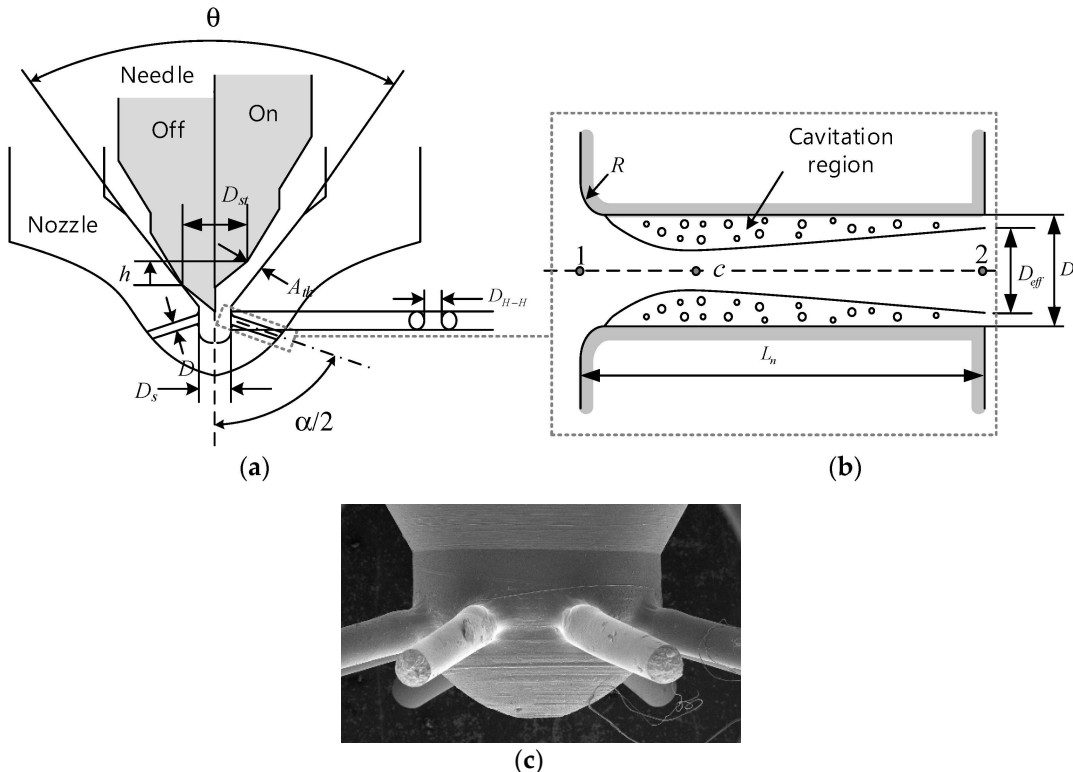

**Figure 1.** Schematic diagram and SEM image of the injector nozzle. (**a**) Nozzle with needle; (**b**) Nozzle flow model [11,12]; (**c**) Scanning Electron Microscope (SEM) image of the nozzle hole.

*2.2. Design of Injector Nozzle Hole*

The nozzle exit flow affects the spray and vaporization of fuel, directly related to engine combustion. The nozzle flow model in Figure 1b is a zero-dimensional model that calculates pressure and velocity at each points of the nozzle flow path. Using the nozzle flow model, the effective mean velocity ($U_{mean}$), effective diameter ($D_{eff}$), and discharge coefficient ($C_d$) can be obtained from the geometric shape of the nozzle and the injected fuel quantity from the experiment [10,11]. In this study, it was modified to predict the discharge coefficient by using inlet injection pressure, instead of the injection quantity obtained through experiments, and nozzle geometry for manufacturing DME injector nozzles.

Figure 2 shows the simplified injection pressure characteristics at the nozzle inlet. In the nozzle flow model, the inlet injection pressure is simply assumed as a sine function considering needle movement.

$$P_{inj} = P_1 = P_2 + C \times \sin(2\pi t/T) \tag{1}$$

where, $P_{inj}$ is the injection pressure (Pa), $P_2$ is the pressure of $5 \times 10^6$ (Pa) in the combustion chamber, C is the desired maximum pressure (Pa) to the combustion chamber pressure, and T/2 is the injection duration of 1 (ms). Thus, at time of $t = 0.5$ (mm), it becomes peak injection pressure of $P_{inj} = P_2 + C$.

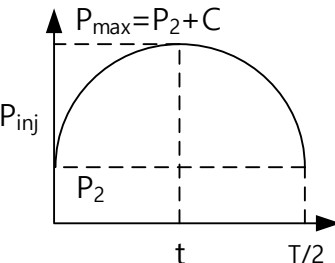

**Figure 2.** The simplified injection pressure characteristics at the nozzle inlet.

If the injection pressure is determined at the nozzle inlet from Equation (1), the initial discharge coefficient is first assumed. Then, the discharge coefficient can be obtained by calculating the equation related to the average flow rate, repeatedly.

$$U_{mean} = C_d \sqrt{\frac{2(P_{inj} - P_2)}{\rho_l}} \tag{2}$$

$$C_d = \frac{1}{\sqrt{1 + f + \frac{L_n}{D} + K_i}} \tag{3}$$

where, $U_{mean}$ is the effective mean velocity (m/s), $C_d$ is the discharge coefficient (-) and $\rho_l$ is the density (kg/m$^3$) of the fuel as diesel or DME. In addition, $f$ is friction loss coefficient (-), $L_n$ is the length of the nozzle (mm), $D$ is the nozzle diameter (mm), and $K_i$ is the entrance loss coefficient (-).

The entrance loss coefficient ($K_i$) is the function of $R/D$, the ratio of the inlet radius ($R$) to the nozzle hole diameter ($D$). The friction loss coefficient ($f$) in Equation (3) can be determined by [11]:

$$f = \max(\frac{64}{Re}, \ 0.316Re^{-0.25}) \tag{4}$$

where, $Re = \nu U_{mean} D$ is Reynolds number in the nozzle hole.

In order to evaluate whether the flow is cavitating and or not in the nozzle, the vena contracta pressure ($P_{vena}$) at point c inside the nozzle in Figure 1b can be obtained from Equation (5) to Equation (7):

$$C_c = \frac{1}{\sqrt{2.6787 - 11.4 \, R/D}} \tag{5}$$

$$U_{vena} = \frac{U_{vena}}{C_c} \tag{6}$$

$$P_{vena} = P_{inj} - \frac{\rho_l}{2}U_{vena}^2 \tag{7}$$

where, $C_c$ is the vena contracta coefficient (-), $P_{vena}$ is the pressure (Pa), and $U_{vena}$ is the vena contracta velocity (m/s) at point c, as shown in Figure 1b.

As the vena contra velocity increases with higher injection pressure, the vena contracta pressure decreases. If pressure ($P_{vena}$) at this point is less than the vapour pressure ($P_{vap}$) of the fuel, cavitation occurs inside the nozzle. The injection pressure and discharge coefficient ($C_d$) can be re-calculated as:

$$P_{inj} = P_{vap} + \frac{\rho_l}{2}U_{vena}^2 \tag{8}$$

$$C_d = C_c \cdot \sqrt{\frac{P_{inj} - P_{vap}}{P_{inj} - P_2}} \tag{9}$$

where, $P_{vap}$ is the vapour pressure of the fuels (Pa) and $P_2$ is the pressure in the combustion chamber (Pa).

Once the flow characteristics inside the nozzle are determined, the effective diameter, the effective velocity, and total injection rate of the nozzle are calculated by:

$$U_{eff} = U_{vena} - \frac{P_{inj} - P_{vap}}{\rho_l U_{mean}} \tag{10}$$

$$A_{eff} = A_{hole} \frac{U_{mean}}{U_{eff}} \tag{11}$$

$$D_{eff} = \sqrt{\frac{4A_{eff}}{\pi}} \tag{12}$$

$$\dot{m}_t = \rho_l \cdot U_{eff} \cdot A_{eff} \cdot N_h \tag{13}$$

where, $U_{eff}$ is the effective velocity (m/s), $A_{eff}$ is the effective area in the nozzle exit (mm$^2$), $D_{eff}$ is the effective diameter in the nozzle exit (mm), and $\dot{m}_t$ is total injection rate (g/s).

In this study, the operating temperatures of both fuels are assumed to be constant at 20 °C.

Figure 3 shows the flow chart of the injector nozzle design based on the nozzle flow model in this work.

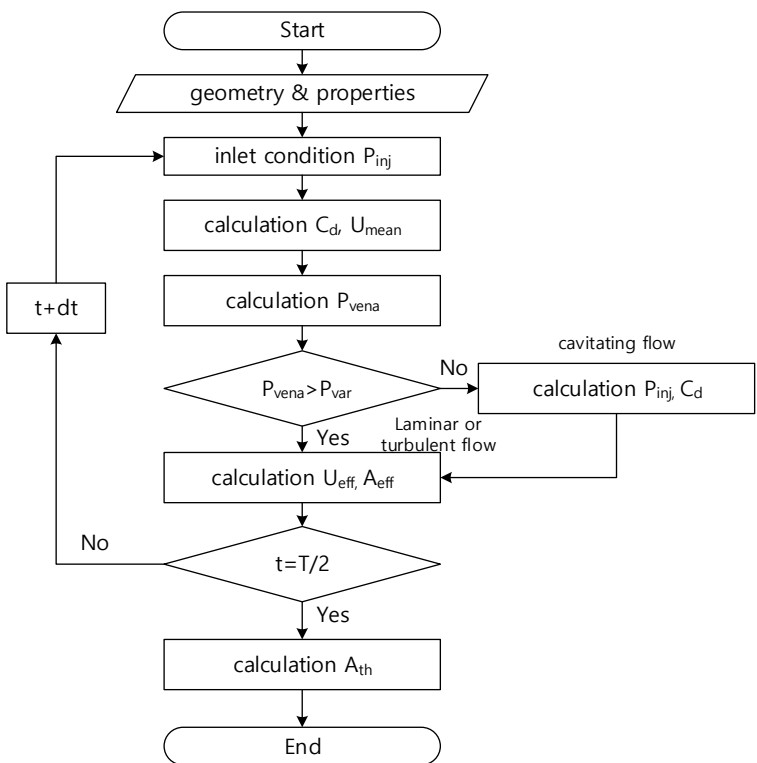

**Figure 3.** Flow chart of the injector nozzle design.

*2.3. Experiment*

The injection rate was measured to validate the nozzle flow model. The injection rate indicates the fuel flow rate over time during the injection period. Therefore, the peak injection rate ($\dot{m}_{max}$) represents the highest injection rate during that injection period. The Bosch tube method was applied for injection rate measurement [13]. It calculates the injection rate by measuring the pressure wave inside the tube when fuel is injected into the tube. Figure 4 shows the experimental setup for injection rate measurement in this study. The fuel-injected from the nozzle flows into the tube through the injector adaptor, as shown in Figure 4b. The pressure transducer is installed near the injector nozzle tip for pressure wave measurement. Figure 5 shows the control volume moving at speed (*c*) with sound

waves when the fuel is injected and flows at speed $u$ inside the tube with constant cross-section ($A$). Applying continuity and momentum equations to this control volume offers the following expressions:

$$\rho_l(c - u)A - (\rho_l + d\rho_l)(c - u - du) = 0 \tag{14}$$

$$(c - u)\rho_l(c - u)A - (c - u - du)(\rho_l + d\rho_l)(c - u - du) = Adp \tag{15}$$

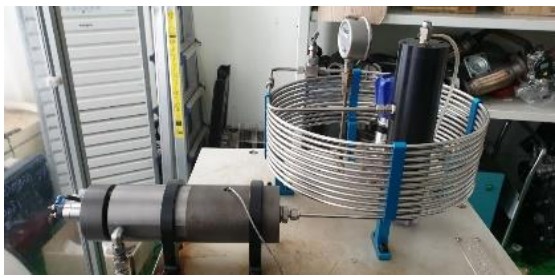
(**a**) Injection rate meter

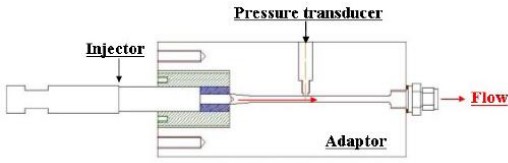
(**b**) Adaptor part in injection rate meter

**Figure 4.** Experimental setup for injection rate measurement.

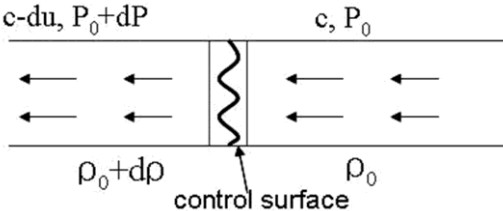

**Figure 5.** Control volume chosen for analysis.

From Equations (14) and (15), Equation (16) can be derived:

$$dp = c\rho_l du \tag{16}$$

The injection rate ($Q$) can be obtained from pressure wave ($p$) by integration of Equation (16) and then substituting it for the injection rate equation:

$$\frac{dQ}{dt} = \frac{A}{\rho_l c}p \tag{17}$$

The injection rate was calculated using the average value of 100 cycles of pressure wave in each test condition. The pressure data for 100 cycles was recorded and averaged with an oscilloscope.

## 3. Results and Discussion

### 3.1. Nozzle Hole Diameter and Needle Lift

Figure 6 shows the peak injection rate ($\dot{m}_{max}$) for injection pressure of the diesel and DME systems. It is based on solenoid injector with 0.14 (mm) nozzle diameter and eight holes in a common rail system. The design requirement for the injection pressure of DME was set to be $P_{max} = 40$ (MPa), while that of diesel was $P_{max} = 160$ (MPa). DME gasifies immediately during injection, due to its low boiling point, even though it is injected as a liquid. Therefore, the high fuel injection pressures, such as $P_{max} = 50$–160 (MPa) used in modern diesel injection systems are not required for DME [5]. The peak injection rates were experimentally measured by the Bosch tube method and numerically predicted by the zero-dimensional nozzle flow model. In the experiment, it was difficult to measure nozzle injection pressure in the nozzle inlet, as shown in Figure 1c; the pressure in the common rail system is regarded as the inlet injection pressure. In the experiment, peak injection rate was calculated using the average

value of 100 cycles from an injection rate at time of 0.5 (ms). The trend predictions for peak injection rate are in good agreement with the experimental results. However, the peak injection rate using the nozzle flow model was found to be overpredicted, compared to those of the experiment results. There are certain assumptions in the nozzle flow model. Because the fuel in the common rail system enters the nozzle inlet through the orifices in the injector, a pressure drop occurs through the orifices. Therefore, the nozzle inlet pressure, which determines the peak injection rate, is low in the experiment. In the nozzle flow model, the inlet injection pressure is assumed as a sine function, as shown in Figure 2. In general, the peak injection rate occurs after T/2. In addition, the nozzle flow model presented in this work did not consider the effect of the flow reduction due to the surface roughness inside the nozzle. Thus, the peak injection rates of the nozzle flow model are higher than those of the experiment. In Figure 6, the peak injection rate is 0.0559 (g/s) for the required maximum pressure in the diesel common rail system, $P_{max} = 160$ (MPa). The peak injection rate is 0.0278 (g/s) with the designed maximum pressure of 50 (MPa) in the DME common rail system, which is 49.7% of the diesel system. In addition, since the lower heating value of DME ($Q_{LHV}$) is 27.6 (MJ/kg) and 65% of diesel, more DME is required in the combustion chamber to obtain the equivalent engine power. To increase the injection rate, injection duration should be expanded. However, it is necessary to secure a peak injection rate since there are limits to increasing injection duration. Therefore, the injector nozzle design should be changed to compensate for insufficient DME heating value when applying DME common rail system of $P_{max} = 50$ (MPa) to a conventional diesel common rail system of $P_{max} = 160$ (MPa).

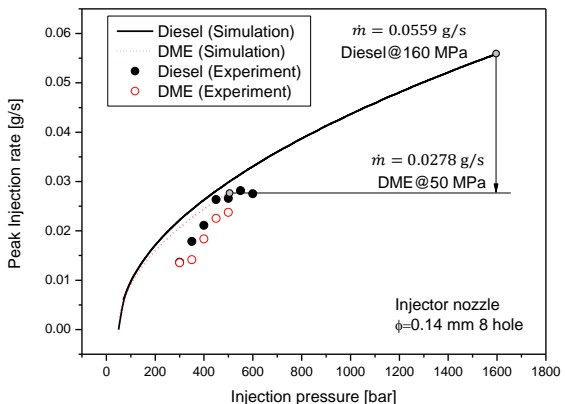

**Figure 6.** Peak injection rate according to injection pressure.

Figure 7 shows the peak injection rate ($\dot{m}$) according to nozzle hole diameter for DME injection pressure of 50 (MPa). To ensure sufficient injection rate in the nozzle, an increase in nozzle diameter is required, as shown in Figure 6. As the diameter of the nozzle increases by 0.2 (mm), the peak injection rate becomes 0.0559 (g/s), which is the same amount of that of 160 (MPa). When the diameter of the nozzle is 0.245 (mm), the peak injection rate is 0.086 (g/s). It is capable of obtaining equivalent engine power considering the lower heating value, as shown in Figure 7. However, as the diameter of the hole increases from 0.14 (mm) to 0.245 (mm), the distance between the holes decreases from 1.64 (mm) to 0.8 (mm). The minimum distance between holes ($D_{H-H}$) has a significant effect on the reliability of the nozzle operating at high pressure; thus, it is recommended to maintain a certain distance.

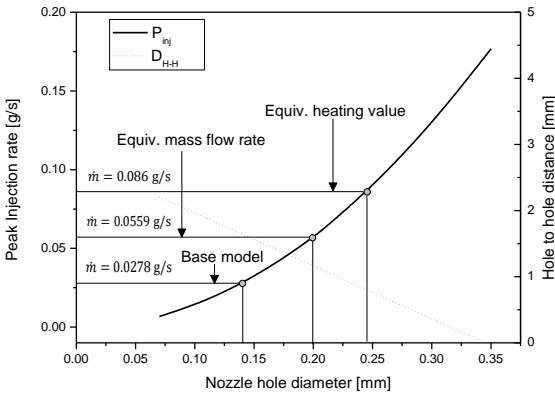

**Figure 7.** Peak injection rate according to nozzle hole diameter for the base injector.

Figure 8 shows the flow area of the seat ($A_{th}$) for the needle lift. It is difficult to obtain the desired amount of DME if the design for the needle seat is not changed, even when the nozzle hole diameter is increased. The flow area of the seat, as shown in Figure 1a, is sufficient to compensate for the overall area of the nozzle due to increase of the nozzle diameter. Meanwhile, there are design limitations to increasing the needle lift in order to obtain a sufficient injection rate of DME. If the needle lift is only increased from 0.1 (mm) to 0.2 (mm), the flow area of the seat will be approximately 0.31 (mm²), which is about 2.1 times larger than that of the base model with 0.1 (mm). The diameter of the needle increased by 1.88 (mm) in order to obtain a larger flow area of the seat. As a result, the flow area of the seat with $h = 0.15$ (mm) and $D_{st} = 1.88$ (mm) was modified to increase by more than four times, compared to the base model, to secure a sufficient flow area.

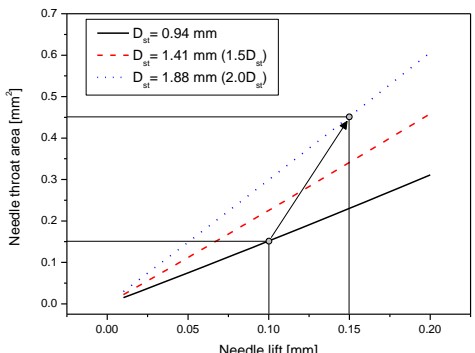

**Figure 8.** Area ratio according to needle lift.

*3.2. Discharge Coefficient and Effective Nozzle Exit Diameter*

Figure 9 shows discharge coefficient ($C_d$) during the injection period with maximum injection pressure of 160 (MPa) for diesel and 50 (MPa) for DME. Typically, the injection rate increases as injection pressure rises. Due to the reduction of friction in the nozzle, the discharge coefficient gradually increases. However, when the pressure at the vena contracta is lower than the vapour pressure of fuels, the cavitation occurs. It reduces the discharge coefficient during injection. Since the injection pressure increases, the cavitation appears for both diesel and DME. The maximum discharge coefficient of DME is $C_d = 0.91633$ and higher than that of diesel. Because the kinematic viscosity ($\nu$) of DME is lower than that of diesel, the flow efficiency of DME in the nozzle is higher. The maximum discharge coefficients also increase according to increase of the diameter of the nozzle hole for DME fuels. As shown in Table 2, the vapour pressure of DME is 550 (kPa), higher than that of diesel, $P_{vap} = 10$ (kPa). Thus, cavitation in DME starts to occur at a lower inlet injection pressure, compared to that of diesel. However, because the maximum injection pressure in DME is lower than that in diesel, the period of cavitation in DME is narrower. Meanwhile, the cavitation period slightly expands due to increase of

the nozzle diameter for DME, as shown in Figure 9. This is because the effect of the increased flow rate in cavitation is greater than suppression of the cavitation due to the increased diameter of the nozzle. Cavitation conditions such as inlet injection pressure and period of cavitation region for diesel and DME are summarized in Table 3.

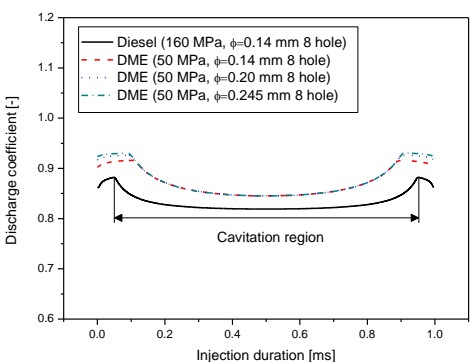

**Figure 9.** Discharge coefficient during injection.

**Table 3.** Prediction of cavitation conditions for diesel and DME.

| Item | Inlet Injection Pressure for Cavitation (MPa) | Period of Cavitation Region (ms) |
|---|---|---|
| Diesel ($P_{max} = 160$ MPa, $\varnothing = 0.14$ mm 8 hole) | 30.21 | 0.896 |
| DME ($P_{max} = 50$ MPa, $\varnothing = 0.14$ mm 8 hole) | 19.98 | 0.784 |
| DME ($P_{max} = 50$ MPa, $\varnothing = 0.20$ mm 8 hole) | 18.91 | 0.800 |
| DME ($P_{max} = 50$ MPa, $\varnothing = 0.245$ mm 8 hole) | 18.37 | 0.808 |

Figure 10 shows the ratio of the effective nozzle exit diameter ($D_{eff}$) under injection period with maximum injection pressure of 160 (MPa) for diesel and 50 (MPa) for DME. The effective nozzle exit diameter is an important design parameter that determines the initial SMD (sauter mean diameter) in combustion analysis with the diameter of the fuel being injected through the nozzle [5]. The ratio of the effective nozzle exit diameter is initially kept at 1.0 and then gradually decreased when cavitation occurred inside the nozzle. The ratio of the effective nozzle exit diameter is 0.905 for diesel, compared to 0.920 in DME at maximum injection pressure condition of $t = 0.5$ (ms). Since the maximum injection pressure of diesel is higher than that of DME, the ratio of the effective nozzle exit diameter reduces for the wider range of injection duration. In the case of DMEs, as the diameter of the nozzle increases, the point of reduction in the ratio of the effective nozzle exit diameter occurs slightly earlier, as is shown in Figure 9.

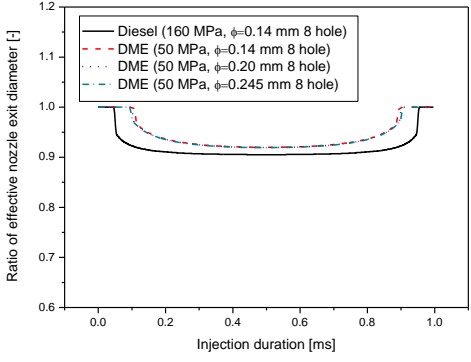

**Figure 10.** The ratio of effective nozzle exit diameter during injection.

## 4. Conclusions

A study using the zero-dimensional nozzle flow model was conducted to design an injector nozzle for the application of DME to conventional diesel common rail system. The injector nozzle design should be changed to compensate for insufficient DME heating value when applying the DME common rail system, $P_{max} = 50$ (MPa), to the conventional diesel common rail system, $P_{max} = 160$ (MPa). The nozzle flow model is such that the flow in the nozzle is modelled as a quasi-steady, zero-dimensional flow, which can be divided and analysed for cavitating flow and non-cavitating flow. The nozzle flow model determines boundary conditions for combustion analysis. A new injector nozzle for DME injection pressure was designed to have the same engine power as the diesel common rail system. In addition, the discharge coefficient and effective diameter of the nozzle exit was analyzed for injection pressure and nozzle geometry of the diesel and DME systems by using the nozzle flow model. Changing fuel from diesel to DME and reducing the injection pressure from 160 (MPa) to 50 (MPa) decreased injection quantity by approximately 49.7%. Therefore, increasing nozzle hole diameter by 0.2 (mm) provides the equivalent mass flow rate as diesel and, when increased by 0.245 (mm), it offers equivalent heating value. In addition, the SAC inner diameter and needle lift was changed by 1.88 (mm) and 0.15 (mm) to ensure overall area of the nozzle. Despite the low injection pressure of DME, the discharge coefficient was found to be higher than that of diesel due to the lower kinematic viscosity and narrowing of period of cavitation region in DME, compared to that of diesel for the duration of injection. The ratio of the effective nozzle exit diameter was initially kept at 1.0 and then gradually decreased when cavitation occurs inside the nozzle.

**Author Contributions:** S.-W.H. played a leading role in writing the paper as a first author. G.-S.L. is the corresponding author and designed paper. H.-C.K. is co-author and helped to write the discussions in the article. Y.-S.S. analysed the data obtained by experiment. All authors have read and approved the final manuscript.

**Funding:** The authors would like to appreciate the Ministry of Trade, Industry and Energy of the Republic of Korea for their financial support ("Demonstration Research Project of Clean Fuel DME Engine for Fine Dust Reduction", Project No. 20182010106370).

**Conflicts of Interest:** The authors declare no conflict of interest.

## Nomenclature

| | |
|---|---|
| $A$ | Cross-section area in Bosch tube method (mm$^2$) |
| $A_{eff}$ | Effective area in the nozzle exit (mm$^2$) |
| $A_{hole}$ | Nozzle hole area (mm$^2$) |
| $\alpha$ | Injection angle (°) |
| $c$ | Vena contracta position (-), sound wave speed (m/s) |
| $C$ | Desired maximum pressure (Pa) |
| $C_c$ | Contraction coefficient (-) |
| $C_d$ | Discharge coefficient (-) |
| $D$ | Nozzle hole diameter (mm) |
| $D_{eff}$ | Effective diameter (mm) |
| $D_{H-H}$ | Hole to hole distance (mm) |
| $D_s$ | Diameter of SAC (mm) |
| $D_{st}$ | Diameter of needle seat (mm) |
| $f$ | Friction coefficient (-) |
| $h$ | Needle lift (mm) |
| $K_i$ | Entrance loss coefficient (-) |
| $L_n$ | Length of nozzle hole (mm) |
| $\dot{m}_{max}$ | Peak injection rate (g/s) |
| $\dot{m}_t$ | Total injection rate (g/s) |
| $N_h$ | Number of hole (-) |
| $P_{inj}$ | Inlet injection pressure (Pa) ($= P_1$) |

| | |
|---|---|
| $P_2$ | Outlet pressure (Pa) |
| $P_{max}$ | Injection pressure at peak injection rate (Pa) |
| $p_{vap}$ | Vapour pressure (Pa) |
| $P_{vena}$ | Vena contracta pressure (Pa) |
| $Q$ | Injection rate (mm$^3$) |
| $Q_{LHV}$ | Lower heating value of fuel (MJ/kg) |
| $R$ | Inlet radius of nozzle (mm) |
| $Re$ | Reynolds number (mm) ($= \nu U_{mean} D$) |
| $\rho_l$ | Density of fuel (kg/m$^3$) |
| $u$ | Control volume speed in Bosch tube method (m/s) |
| $U_{vena}$ | Vena contracta velocity (m/s) |
| $U_{eff}$ | Effective velocity (m/s) |
| $\theta$ | Seat angle (°) |
| $T/2$ | Injection duration (ms) (=1 ms) |

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
