# Peer review of "Study on the Common Rail Type Injector Nozzle Design Based on the Nozzle Flow Model"

_applsci, doi:10.3390/app10020549_

Round 1

Reviewer 1 Report

Most citations of patents are not available in a short period of review.
 So I could not fully understand the symbols used in eqations.
Therefore, I could not judge the results were validity or not.
I think that Authors should be added a calculation flowchart and descroiption.

(1) The nozzle flow model in details was difficult to understand, because I can not obtain these references [6] and [7].
It seems that the value of each parameter needs to be shown more clearly.
Especially, basic physical property used in simulation should be listed.
Also, the nozzle conditions for calculation should also be indicated.

(2) What does the injection angle mean?
The angle in the text (page2 total line49)does not match the angle in the table1.

(3) Author wrote that in section 2.3 as "Bosch tube method was applied for injection-----".
 How did you measure the pressure wave and how did you calculate the injection rate from the pressure?
Please add the explanation and the eqations.

(4) In Figure 3, how did authors obtain the simulation values? If possible, please add  calculation flowchart.

(5) The characters used in Eq.(12) are ambiguous. Please add the explanation in text.

(6) What is difiniton of the "peak injection rate"?
The explanation were written in page4. (total line number 125-126 )
Which part in figure 1 was connected to "the "tube" ?
Please explain more clearly.
.
(7) The sentence  is incorrect (P3 total line number 97).

(9) 4. Conclusions
nozzle halll ->nozzle hole (p6 total line number 193)

Author Response

We appreciate the reviewer for the constructive comments. The suggestions offered by the reviewer have been helpful. Each comment has been considered carefully. Responses to the comments are as follows. Please note that page and line numbers refer to the revised manuscript.

Reviewer 2 Report

Comments for improvement of the reviewed manuscript:

(i) The main question consists in correspondence of the content of the article and the purpose of the journal. I think Authors should improve Introduction using more References in this journal.

(ii) Justify the choice of ranges for varying the parameters of the processes studied.

(iii) Provide and clarify the errors in the measurements. Confidence intervals or experimental points should be shown in the main figures.

(iv) Please explain each of used basic assumptions with references analysis.

(v) Graphical abstract has to add to article.

(vi) Please add Nomenclature section.

(vii) Expand the physical analysis of the obtained research results with comparison of experimental and theoretical results from other authors. A qualitative comparison is advisable.

(viii) Confidence Intervals should be add to Figures and discussed it.

Author Response

(The authors gave the same response as above.)

Round 2

Reviewer 1 Report

The author has responded to the questions properly,

has corrections and additions against my comments.

So I think that this paper would be acceptable for publication.

Thank you.

Reviewer 2 Report

Article can be accepted.